# Effective Layer Pruning Through Similarity Metric Perspective

**Ian Pons** [1]  **Bruno Yamamoto** [1]  **Anna H. Reali Costa** [1]  **Artur Jordao** [1]

## Abstract

Deep neural networks have been the predominant paradigm in machine learning for solving cognitive tasks. Such models, however, are restricted by a high computational overhead, limiting their applicability and hindering advancements in the field. Extensive research demonstrated that pruning structures from these models is a straightforward approach to reducing network complexity. In this direction, most efforts focus on removing weights or filters. Studies have also been devoted to layer pruning as it promotes superior computational gains. However, layer pruning often hurts the network predictive ability (i.e., accuracy) at high compression rates. This work introduces an effective layer-pruning strategy that meets all underlying properties pursued by pruning methods. Our method estimates the relative importance of a layer using the Centered Kernel Alignment (CKA) metric, employed to measure the similarity between the representations of the unpruned model and a candidate layer for pruning. We confirm the effectiveness of our method on standard architectures and benchmarks, in which it outperforms existing layer-pruning strategies and other state-of-the-art pruning techniques. Particularly, we remove more than 75% of computation while improving predictive ability. At higher compression regimes, our method exhibits negligible accuracy drop, while other methods notably deteriorate model accuracy. Apart from these benefits, our pruned models exhibit robustness to adversarial and out-of-distribution samples.

## 1. Introduction

It is well known that deep neural networks are capable of obtaining remarkable results in various cognitive fields, of-

[1]Escola Politécnica, Universidade de São Paulo. Correspondence to: Ian Pons <ian.pons@usp.br >.

Accepted to the Workshop on Advancing Neural Network Training at International Conference on Machine Learning (WANT@ICML 2024).

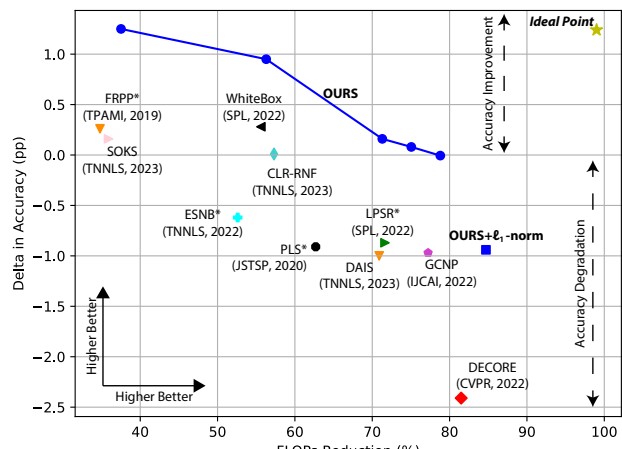

Figure 1. Comparison with state-of-the-art on the popular ResNet56 + CIFAR-10 setting. (Here, for illustration purposes, we abuse notation and bound the ideal point close to two percentage points (pp); however, it may be higher). Overall, our method obtains the best compromises between accuracy and computational reduction (estimated by Floating Point Operations – FLOPs). Specifically, our method dominates existing layer pruning methods (indicated by symbol *) by a remarkable margin. Compared to state-of-the-art pruning techniques, our method removes more than 75% of FLOPs without hurting accuracy (sometimes improving it). Other methods, however, degrade accuracy when operating at these high FLOP reduction regimes. Since our method is orthogonal to modern structured filter pruning, we can combine them to achieve even higher computation gains (e.g., Ours + $\ell_1$-norm). The behavior shown in this figure is consistent across other benchmarks and architectures.

ten outperforming humans from image recognition to complex games such as chess and go (Silver et al., 2016; Han et al., 2023). However, this performance comes with high computational cost and storage demand. Advances in the foundation model paradigm – large deep learning models trained on a broad range of data with the capacity to transfer its knowledge to unseen (downstream) tasks – have further intensified the resource-intensive nature of the field, as large models play a crucial role in transferring knowledge to downstream tasks (Bommasani et al., 2021; Amatriain et al., 2023).

To mitigate the above issues, compression techniques are

becoming more popular due to their positive results in improving the resource demand of deep models (He & Xiao, 2023; Xu & McAuley, 2023; Wang et al., 2022). Among the most promising techniques, pruning emerges as a straightforward approach capable of enhancing different performance metrics such as Floating Point Operations (FLOPs), memory consumption and number of parameters. At the heart of a pruning technique lies the task of accurately estimating the importance of structures that compose a model and subsequently removing the least important ones.

Studies classify pruning into unstructured and structured categories (He & Xiao, 2023; Xu & McAuley, 2023; Wang et al., 2022). The former removes individual connections (weights), while the latter focuses on eliminating entire structures such as filters and layers. Despite achieving high compression rates, unstructured approaches promote theoretical inference speed-up only, requiring specific hardware to handle sparse matrix computations to obtain practical speed-up (Wang et al., 2022; Xu & McAuley, 2023). On the other hand, structured pruning facilitates practical acceleration without any hardware/software constraints. Furthermore, its advantages extend beyond computational gains: it acts as a regularization mechanism that improves generalization and robustness (Zhao & Wressnegger, 2023; Bair et al., 2024).

Most efforts on structured pruning strategies focus on eliminating small structures and are often optimized for standard metrics such as FLOP and parameter reduction (He & Xiao, 2023; Xu & McAuley, 2023). Unfortunately, recent studies suggest that these metrics may correlate weakly with inference time (Shen et al., 2022; Dehghani et al., 2022; Vasu et al., 2023). Nevertheless, layer pruning reduces network depth, which directly addresses model latency while also providing all the benefits of filter pruning, such as FLOP and parameter reduction, without specialized software or hardware (Jordao et al., 2020; Zhang et al., 2022; Zhou et al., 2022). The idea behind pruning layers is not novel and dates back to 2016 (Veit et al., 2016; Huang et al., 2016). Efforts in this direction, however, either apply simple filter criteria and combine (e.g., average) the scores to compose the importance of a layer (Jordao et al., 2020; Zhang et al., 2022) or solve a (computationally expensive) multi objective optimization (Zhou et al., 2022). Therefore, one of the challenges in layer removal is developing a criterion capable of accurately ranking the importance of all layers. It turns out that existing criteria operate well on small structures but may be inadequate when applied to large ones, primarily because of varying magnitudes (i.e., $\ell_1$-norm and its variations) exhibited by layers (Zhang et al., 2022; Jordao et al., 2023). Additionally, recent studies highlight that different layers play a distinct role in the expressive power and training dynamics of deep models (Zhang et al., 2022; Masarczyk et al., 2023; Chen et al., 2023). Such factors suggest that simple criteria are unable to characterize all these underlying properties exhibited by layers. Lastly, a layer-pruning method must inherit a fundamental requirement of pruning techniques: remove structures without significantly compromising predictive ability.

To meet the aforementioned requirements and achieve computational-friendly models, we propose a novel layer pruning method. Our method relies on the hypothesis that similar representations between a dense (unpruned) network and its optimal sparse (pruned) candidate indicate lower relative importance. Existing evidence support this idea, revealing that layers share similar representations (Zhang et al., 2022; Masarczyk et al., 2023). By eliminating unimportant layers, we can preserve predictive capability and reduce computational demand.

For this purpose, we employ Centered Kernel Alignment (CKA) due to its effectiveness and flexibility in measuring similarity between two networks (Kornblith et al., 2019; Nguyen et al., 2021; 2022; Masarczyk et al., 2023). Leveraging CKA, the overall of our method is the following. Given a dense network, we first extract its representation from some input examples. Here, representation refers to the feature maps of the layer just before the classification layer. Then, we create a temporary pruned model by removing a candidate layer. Building upon previous works (Veit et al., 2016; Zhang et al., 2022), at this step, we avoid any fine-tuning or parameter adjustment, since modern architectures are robust to single layer removal and perturbations. Afterward, for each temporary model ($n$ layers, $n$ candidates), we compare their representations with the original network using CKA. Finally, we select the temporary network that yields the highest similarity, thus removing the unimportant layer.

**Contributions**. We highlight the following key contributions. First, we propose a novel pruning criterion that leverages an effective similarity representation metric: CKA. To the best of our knowledge, we are the first to explore CKA as a pruning criterion, as previous works widely employ it for comparing network representations. Powered by this criterion, we develop a layer-pruning method that removes entire layers from neural networks without compromising predictive ability. Such a result is possible since our criterion identifies unimportant layers – layers that, when removed, preserve similarity regarding the original model. In particular, we aggressively prune ResNet56/110 and ResNet50, encouraging them to converge towards ResNet20 in terms of depth and computational cost but preserving their original accuracy. Second, unlike most existing layer-pruning criteria that fail to capture underlying properties of layers, our method effectively assigns layer importance and thus prevents model collapse. Besides, it is efficient and scales linearly as a function of the network depth. Third, we outperform state-of-the-art pruning methods by a notable

margin (see Figure 1). We believe our results open new opportunities to prune through the lens of emerging similarities metrics (Williams et al., 2021; Duong et al., 2023) and encourage further efforts on layer pruning.

Through extensive experiments on standard architectures and benchmarks, we demonstrate that our method outperforms state-of-the-art pruning approaches. Specifically, it surpasses existing layer-pruning strategies by a large margin. In particular, as we increase the levels of FLOP reduction, most layer-pruning methods fail to preserve accuracy, even when equipped with additional techniques such as knowledge distillation (Chen & Zhao, 2019; Zhou et al., 2022). Our method, on the other hand, successfully maintains accuracy while eliminating more than $75\%$ of FLOPs. At a FLOP reduction exceeding $80\%$, our method exhibits negligible accuracy drop, whereas other state-of-the-art techniques are unable to achieve similar performance without compromising accuracy roughly $2\times$ more. We also demonstrate that our method preserves generalization in out-of-distribution and adversarial robustness scenarios, which is crucial for deploying pruned models in security-critical applications such as autonomous driving. In terms of Green AI (Lacoste et al., 2019; Strubell et al., 2019; Faiz et al., 2024), our method reduces the carbon emissions required in the training/fine-tuning phase by up to $80.85\%$, representing an important step towards sustainable AI. Source code and models available at: *link after review*.

## 2. Related Work

The main (and most challenging) task of pruning is to estimate the relative importance of a given structure to differentiate between those essential for predictive ability and the less important ones. A popular criterion focuses on the magnitude of weights, namely $\ell_p$-norm. Researchers extensively explore these criteria in the context of the lottery ticket hypothesis and pruning at initialization (Wang et al., 2022). Despite their simplicity, previous works pointed out pitfalls in these criteria (Zhang et al., 2022; Huang et al., 2021; He et al., 2019). For example, Huang et al. (Huang et al., 2021) argued that constraining the analysis to surrounding structures, as $\ell_1$-norm does, incurs a low variance of importance scores, hindering unimportant structure search. Furthermore, comparing norms across layers becomes impractical, as different layers exhibit distinct magnitudes, posing a challenge for global pruning (i.e., ranking all structures at once) (Zhang et al., 2022; Jordao et al., 2023). These issues have motivated efforts towards more elaborate criteria (Shen et al., 2022). Taking the work by Lin et al. (2020) as an example, the authors proposed estimating filter importance based on the rank of its feature maps. Pruning strategies that leverage information from feature maps (thus involving data forwarding through the network) are named *data-driven*

techniques. Since we measure similarity from feature maps, our method belongs to this category of pruning.

Shen et al. (2022) measure filter importance based on the Taylor expansion of the loss change. Importantly, they highlighted the relevance of focusing on latency instead of standard metrics such as FLOPs. To tackle this challenge, the authors transformed the objective of maximizing accuracy within a given latency budget into a resource allocation optimization problem, then solved it using the Knapsack paradigm. In an alternative line of research, studies have demonstrated that standard performance metrics may correlate weakly with inference time (Dehghani et al., 2022; Vasu et al., 2023). Aligned with these efforts, we demonstrate that our method achieves notable latency improvements and other computational benefits. Differently from Shen et al. (2022), we address the accuracy/latency trade-off without solving any optimization problem. This is possible because layer pruning reduces network depth, directly translating into latency improvement, and our criterion accurately identifies unimportant layers that preserve accuracy, enabling us to achieve higher FLOP reductions while maintaining accuracy simultaneously.

According to existing works (Xu & McAuley, 2023; He & Xiao, 2023), most efforts have been devoted to filter pruning techniques. In contrast to this family of methods that may exhibit bias toward specific metrics like FLOPs or parameters (Dehghani et al., 2022; Vasu et al., 2023), layer pruning achieves performance gains across all computational metrics (Chen & Zhao, 2019; Jordao et al., 2020; Zhou et al., 2022). In this direction, Chen et al. (2019) proposed learning classifiers using features from prunable layers to assign their importance. Following this modeling, layer importance relies on the performance of classifiers. Similar to ours, the criterion by Chen et al. (2019) is layer-specific; however, our criterion focuses on similarity representations through CKA, which we reveal to be more effective. More recently, the work by Zhang and Liu (2022) disconnects residual mapping and estimates its effect using Taylor expansion. Zhou et al. (2022) proposed an evolutionary-based approach, using the weights distribution as one of the inputs for creating the initial population of candidate pruned networks. It is worth mentioning that the methods by Zhou et al. (2022) and Chen et al. (2019) require knowledge distillation to recover accuracy from the pruned models, while our method relies on straightforward fine-tuning rounds. Such observations suggest that our CKA criterion is more precise than the previously mentioned strategies for selecting layers.

Apart from pruning, efforts have also been devoted to understanding the role of layers in the expressive power and training dynamics of the models (Zhang et al., 2022; Masarczyk et al., 2023; Chen et al., 2023). For example, Masarczyk et al. (2023) suggested that the layers of deep networks split

into two distinct groups. The initial layers have linearly separable representations, and the subsequent layers, or the tunnel, have less impact on the performance, compressing the already learned representations. This behavior, named *Tunnel Effect*, emerges at the early stages of the training process and corroborates with the notion of redundancy in overparameterized models. Additionally, their work argued that the tunnel is responsible for the performance degradation in out-of-distribution (OOD) samples. We show that our layer pruning method preserves OOD generalization, indicating that its degradation is not restricted to tunnel layers. In summary, we believe our work contributes to these efforts by demonstrating that unimportant layers can be effectively identified and removed without compromising the expressive power of the model and its training dynamics.

## 3. Preliminaries and Proposed Method

**Problem Statement.** According to previous works (Veit et al., 2016; Huang et al., 2016; Dong et al., 2021), residual-based architectures enable the information flow (i.e., the representation) to take different paths through the network. Thereby, layers may not always strongly depend on each other, reinforcing the idea of redundancy in this type of structure, which suggests the possibility of removing layers without compromising the network representation. It follows that a subset of layers plays a crucial role in the network performance (Zhang et al., 2022; Masarczyk et al., 2023; Chen et al., 2023). Upon this evidence, our problem becomes identifying and removing unimportant layers, preserving the representation capacity of the model, and avoiding network collapse. Formally, given a network $\mathcal{N}$ composed of a layer set $L$, our goal is to remove certain layers to produce a shallower network $\mathcal{N}'$ composed by $L'$, where $|L'| \ll |L|$ and the accuracy of $\mathcal{N}'$ is as close as possible (ideally better) than its unpruned version $\mathcal{N}$.

Naively, one could estimate optimal layers to prune by iterating over all possible candidates, removing one at a time, fine-tuning the model, and selecting the candidate that exhibits the lowest performance degradation. However, this approach becomes computationally expensive as the network depth increases, hence it is unfeasible for most modern architectures and large scale datasets.

Following previous layer-pruning works (Chen & Zhao, 2019; Zhou et al., 2022; Zhang & Liu, 2022), we indeed remove building blocks (set of layers) instead of just individual layers. Throughout the text, however, we opt to use the term *layer pruning* rather than *block pruning* to maintain simplicity and clarity. We provide technical details involving layer pruning in Appendix 6.1.

**Definitions.** Consider $X$ and $Y$ a set of training samples (e.g., images) and their respective class labels. Let $\mathcal{N}$ be a

dense (unpruned) network trained using $X$ and $Y$ (i.e., the traditional supervised paradigm). Consider $M(\cdot, X)$ as a function that extracts the representation of a network from the samples $X$. Following Xu et al. (Xu & McAuley, 2023), $M$ extracts the feature maps from the layer immediately preceding the classification layer of the network. It is worth mentioning that $M$ does not take into account the labels $Y$. Let $l_i \in L$ be the candidate layers (i.e., layers the pruning can eliminate) and, finally, define $\mathcal{N}_{l_i}$ as a pruned candidate network yielded by removing the layer $l_i$ from $\mathcal{N}$.

**Proposed Criterion.** For each $l_i \in L$, we obtain $\mathcal{N}_{l_i}$ w.r.t the previous definition, and apply $M(\mathcal{N}_{l_i}, X)$ to extract its representation, denoted by $R_{l_i}$. Define $s(\cdot, \cdot)$ as our CKA criterion which takes $R$ and $R_{l_i}$, where $R \leftarrow M(\mathcal{N}, X)$ (i.e., the original representation), and outputs the score (importance) of $l_i$. Following Kornblith et al. (2019), we compute CKA in terms of

$$CKA(R, R_{l_i}) = \frac{HSIC(R, R_{l_i})}{\sqrt{HSIC(R, R)HSIC(R_{l_i}, R_{l_i})}}, \quad (1)$$

where HSIC is the Hilbert-Schmidt Independence Criterion (Gretton et al., 2005). Due to space constraints, we refer interested readers to the works by Kornblith et al. (2019) and Nguyen et al. (2021; 2022) for additional information.

It follows from Equation 1 that $CKA(R, R_{l_i}) \in [0, 1]$, where a value of 1 indicates identical feature maps (i.e., the highest similarity preservation). However, an intuitive practice is to remove the lowest-scoring candidate layer. Therefore, we adjust the score in terms of $s(R, R_{l_i}) = 1 - CKA(R, R_{l_i})$, ensuring that lower scores are assigned to layers yielding more similar representations.

Algorithm 1 summarizes the process above. From it, we highlight the following points. First, after estimating the importance of all candidate layers, we indeed remove the lowest scoring one. Second, representation extractions employ the same set $X$. Finally, the construction of $N_{l_i}$ does not involve any fine-tuning.

---

**Algorithm 1** Layer Pruning using our CKA criterion

    **Input:** Trained Neural Network $\mathcal{N}$, Candidate Layers $l_i \in L$
    Training samples $X$
    **Output:** Pruned Version of $\mathcal{N}$
1:  $R \leftarrow M(\mathcal{N}, X)$
2:  **for** $i \leftarrow 1$ **to** $|L|$ **do**
3:     $\mathcal{N}_{l_i} \leftarrow \mathcal{N} \setminus l_i$ ▷ Removes layer $l_i$ from $\mathcal{N}$
4:     $R_i \leftarrow M(\mathcal{N}_{l_i}, X)$ ▷ Representation extraction of $\mathcal{N}_{l_i}$
5:     $S \leftarrow S \cup s(R, R_{l_i})$
6:  **end for**
7:  $j \leftarrow argmin(S)$ ▷ Index of lowest score in $S$
8:  $\mathcal{N} \leftarrow \mathcal{N}_{l_j}$ ▷ $\mathcal{N}$ becomes its pruned version
9:  Update $\mathcal{N}$ via standard fine-tuning on $X$

---

A commonly explored approach in prior studies involves iteratively repeating the pruning and fine-tuning process. We follow such practice by iterating Algorithm 1, where the input to the next iteration is the fine-tuned $\mathcal{N}$ (see line 1 in Algorithm 1). We report the iteration number (hence, the number of removed layers) as a subscript; for example, $CKA_i$ means we perform $i$ iterations to obtain the corresponding pruned model. Note that, for a single iteration, our method scales linearly w.r.t the number of layers, implying an $O(|L|)$ complexity.

# 4. Experiments

**Experimental Setup.** We conduct experiments on CIFAR-10, CIFAR-100 and ImageNet using different versions of the ResNet architecture (He et al., 2016). Such settings are a common choice for general compression/acceleration studies (Chen & Zhao, 2019; Jordao et al., 2020; Zhang et al., 2022; Zhou et al., 2022; He & Xiao, 2023). Throughout both training and fine-tuning phases, we apply random crop and horizontal random flip as data augmentation (He et al., 2016). We choose this simple setup to highlight the genuine advantages of our method. In Appendices 6.3 and 6.4, we also conduct experiments on MobileNetV2 and Transformer architectures.

To compare the predictive ability of the unpruned models with their pruned counterparts, we follow common practices and report the difference between accuracies (He & Xiao, 2023). In this metric, negative and positive values indicate a decrease and an improvement in accuracy (in percentage points – pp), respectively.

**The Effect of Layer Pruning on Efficiency.** Our point of start is illustrating the advantages of layer over filter pruning, as the latter is the most popular family of methods that yield gains without requiring specific hardware (Shen et al., 2022; He & Xiao, 2023).

According to recent studies (Dehghani et al., 2022; Vasu et al., 2023), standard metrics such as FLOPs and parameters, when singly employed, may overlook model efficiency. Therefore, we begin our discussion by considering latency – the time for forwarding a sample (or a set) through the network. To do so, we follow the same process as Jordao et al. (2023), which creates two pruned networks: one obtained through layer removal and the other from filters, aiming for both models to have a similar number of neurons (filters). This procedure makes possible a fair comparison in terms of latency performance.

Iteratively repeating this process yields models with varying numbers of filters removed, from which we measure their average latency across 30 runs by forwarding $10K$ samples and report the speed-up obtained from the pruning process with respect to the original (unpruned) model.

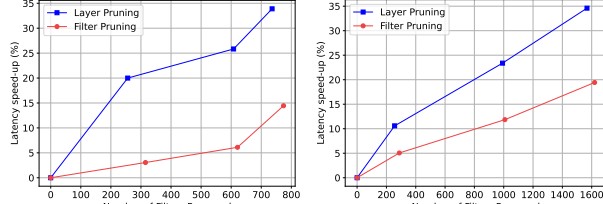

*Figure 2.* Relationship between the number of filters removed (x-axis) and latency speed-up (y-axis) for models obtained from filter and layer pruning. Importantly, such a comparison is possible because when pruning removes layers, it eliminates all filters from that layer. Left and right plots stand for ResNet56 and ResNet110, respectively. Overall, layer pruning notably promotes higher speed-up than filter pruning.

Figure 2 shows the results. It follows that layer pruning yields a higher speed-up than filter removal. For example, in ResNet110, with both methods eliminating around a thousand filters, layer pruning achieves an 11 pp speedup over filter pruning. This advantage persists even when removing approximately $1,600$ filters, underscoring the effectiveness of removing layers for network acceleration. Such gains have motivated previous efforts on layer removal (Jordao et al., 2020; Zhang & Liu, 2022; Zhou et al., 2022).

**Effectiveness of the Proposed CKA Criterion.** Informed by the previous findings, we now turn our attention to evaluating the effectiveness of the proposed CKA criterion in assigning layer importance. For this purpose, we take into account representative layer pruning techniques (Jordao et al., 2020; Zhang et al., 2022; Zhou et al., 2022; Chen & Zhao, 2019). It is worth mentioning that we exclude works on dynamic inference since they belong to a different category of compression and acceleration techniques (Han et al., 2022).

Table 1 summarizes the results. According to this table, our method outperforms existing techniques by a large margin. On ResNet56, compared to the best strategy in terms of delta in accuracy, LPSR (Zhang & Liu, 2022), our method outperforms it by more than $0.76$ pp while exhibiting better FLOP gains. Regarding FLOP reduction, the best method underperforms ours by $3.4$ pp. This behavior is prevalent in ResNet110 and ResNet50 (on ImageNet). Notably, we reduce around $2\times$ more FLOPs than other criteria while obtaining an improvement in accuracy.

The reason for these remarkable results is that our method carefully selects which layers to eliminate. For example, Jordao et al. (2020) and Zang et al. (2022) compute scores for layers by aggregating the sum of scores from the individual filters that compose a layer. Table 1 suggests that this aggregating scheme may be inappropriate. This finding concurs with the observations made by Masarczyk et al. (2023),

*Table 1.* Comparison with state-of-the-art layer-pruning methods. The symbols (+) and (- ) denote increase and decrease in accuracy regarding the original (unpruned) network, respectively. We highlight the best results in bold.

|  | Method | $\Delta$ Acc. | FLOPs (%) |
|---|---|---|---|
|  | PLS (Jordao et al., 2020) | (–) 0.98 | 30.00 |
|  | FRPP (Chen & Zhao, 2019) | (+) 0.26 | 34.80 |
|  | ESNB (Zhou et al., 2022) | (–) 0.62 | 52.60 |
| ResNet56 | LPSR (Zhang & Liu, 2022) | (+) 0.19 | 52.75 |
| on | CKA$_{15}$ (ours) | **(+) 0.95** | **56.29** |
| CIFAR10 | PLS (Jordao et al., 2020) | (–) 0.91 | 62.69 |
|  | LPSR (Zhang & Liu, 2022) | (–) 0.87 | 71.65 |
|  | CKA$_{19}$ (ours) | **(+) 0.16** | 71.30 |
|  | CKA$_{20}$ (ours) | (+) 0.08 | **75.05** |
| ResNet110 | ESNB (Zhou et al., 2022) | (+) 1.15 | 29.89 |
| on | PLS (Jordao et al., 2020) | (+) 0.06 | 37.73 |
| CIFAR10 | CKA$_{27}$ (Ours) | **(+) 1.16** | 50.33 |
|  | CKA$_{36}$ (Ours) | (+) 0.80 | **67.10** |
| ResNet50 | LPSR (Zhang & Liu, 2022) | (-) 1.38 | 37.38 |
| on | CKA$_6$ (Ours) | **(-) 0.18** | **39.62** |
| ImageNet | PLS (Jordao et al., 2020) | (-) 0.67 | 45.28 |
|  | CKA$_7$ (Ours) | **(-) 0.90** | 45.28 |

*Table 2.* Comparison with state-of-the-art pruning methods on CIFAR-10. For each level of FLOP reduction (%), we highlight the best results in bold.

|  | Method | $\Delta$ Acc. | FLOPs (%) |
|---|---|---|---|
|  | DECORE (Alwani et al., 2022) (CVPR, 2022) | +0.08 | 26.30 |
|  | HALP (Shen et al., 2022) (NeurIPS, 2022) | +0.03 | 33.72 |
|  | SOKS (Liu et al., 2023) (TNNLS, 2023) | +0.16 | 35.91 |
|  | **CKA$_{10}$ (ours)** | **+1.25** | **37.52** |
|  | GKP-TMI (Zhong et al., 2022) (ICLR, 2022) | +0.22 | 43.23 |
|  | GCNP (Jiang et al., 2022) (IJCAI, 2022) | +0.13 | 48.31 |
|  | **CKA$_{13}$ (ours)** | **+0.86** | **48.78** |
|  | GNN-RL (Yu et al., 2022) (ICML, 2022) | +0.10 | 54.00 |
|  | RL-MCTS (Wang & Li, 2022) (WACV, 2022) | +0.36 | 55.00 |
| ResNet56 | WhiteBox (Zhang et al., 2023) (TNNLS, 2023) | +0.28 | 55.60 |
|  | CLR-RNF (Lin et al., 2023) (TNNLS, 2023) | + 0.01 | 57.30 |
|  | **CKA$_{16}$ (ours)** | **+0.78** | **60.04** |
|  | DAIS (Guan et al., 2023) (TNNLS, 2023) | -1.00 | 70.90 |
|  | HRank (Lin et al., 2020) (CVPR, 2020) | -2.54 | 74.09 |
|  | GCNP (Jiang et al., 2022) (IJCAI, 2022) | -0.97 | 77.22 |
|  | **CKA$_{20}$ (ours)** | **+0.08** | 75.05 |
|  | **CKA$_{21}$ (ours)** | -0.66 | **78.80** |
|  | DECORE (Alwani et al., 2022) (CVPR, 2022) | -2.41 | 81.50 |
|  | **CKA$_{21}$ (ours) + $\ell_1$** | **-0.94** | **84.70** |
|  | DECORE (Alwani et al., 2022) (CVPR, 2022) | +0.38 | 35.43 |
|  | HRank (Lin et al., 2020) (CVPR, 2020) | +0.73 | 41.20 |
|  | GKP-TMI (Zhong et al., 2022) (ICLR, 2022) | +0.64 | 43.31 |
|  | **CKA$_{24}$ (ours)** | **+1.37** | **44.73** |
|  | DAIS (Guan et al., 2023) (TNNLS, 2023) | -0.60 | 60.00 |
|  | DECORE (Alwani et al., 2022) (CVPR, 2022) | 0.00 | 61.78 |
|  | **CKA$_{35}$ (ours)** | **+0.89** | **65.23** |
|  | EPruner (Lin et al., 2022) (TNNLS, 2022) | +0.12 | 65.91 |
| ResNet110 | CRL-RNF (Lin et al., 2023) (TNNLS, 2023) | +0.14 | 66.00 |
|  | WhiteBox (Zhang et al., 2023) (TNNLS, 2023) | +0.62 | 66.00 |
|  | CCEP (Shang et al., 2022) (IJCAI, 2022) | -0.22 | 67.09 |
|  | **CKA$_{36}$ (ours)** | **+0.80** | 67.10 |
|  | HRank (Lin et al., 2020) (CVPR, 2020) | -0.85 | 68.64 |
|  | **CKA$_{38}$ (ours)** | **+0.59** | **70.83** |
|  | DECORE (Alwani et al., 2022) (CVPR, 2022) | -0.79 | 76.92 |
|  | **CKA$_{41}$ (ours)** | **+0.23** | 76.42 |
|  | **CKA$_{47}$ (ours)** | -0.41 | **87.61** |

where the authors argued that aggregating all features of a layer to compose its final representation is suboptimal, particularly for transfer learning.

In terms of computational cost, compared to Zhou et al. (2022), our method is more cost-friendly. It turns out that this approach solves the score assignment problem through an evolutionary algorithm. Therefore, their method scales expensively as the depth (i.e., $|L|$) increases. On the other hand, to prune a model with $|L|$ layers our approach requires $|L|$ forwards and CKA comparisons, scaling linearly (see Algorithm 1). The method by Jordao et al. (2020) is also linear w.r.t the number of layers, however, it is unable to prune a layer from any region of the network. Specifically, to eliminate a layer $i$, their method requires the removal of all subsequent layers $j$ where $i < j < |L|$.

The previous evidence corroborates the suitability of our criterion for selecting unimportant layers compared to existing state-of-the-art layer pruning methods. Importantly, the discussion above confirms our hypothesis that similar representations between a dense (unpruned) network and its optimal pruning candidate indicate lower relative importance.

**Comparison with the State of the Art.** The previous experiments shed light on the benefits of layer pruning and the effectiveness of our criterion for selecting layers to remove. We now compare our method with general state-of-the-art pruning techniques. For this purpose, we evaluate our method against the most recent and top-performing techniques mainly based on the survey by He et al. (2023). More specifically, we consider methods capable of achieving notable FLOP reduction with negligible accuracy drop. For

a fair comparison, we report the results of each method according to the original paper.

Table 2 shows the results on CIFAR-10 for ResNet56/110. On these architectures, our method outperforms state-of-the-art techniques by removing more FLOPs and achieving the best delta in accuracy. For example, in Table 2 (left), within comparable FLOP reduction regimes, we outperform state-of-the-art methods by a margin starting at approximately 0.4 pp (compared to RL-MCTS (Wang & Li, 2022)) and reaching up to more than 2.5 pp (compared to HRank (Lin et al., 2020)).

Table 2 poses an interesting behavior: at high FLOP reduction levels (i.e., above 70%), all methods fail to preserve accuracy. In contrast, our method removes more than 75% of FLOPs with no accuracy drop. Most cases, our method promotes predictive ability improvements. This benefit is expected, as layer pruning (and its variations) acts as a form of regularization (Huang et al., 2016; Han et al., 2022). Table 2 highlights this behavior in other pruning techniques, but unlike ours, exhibited only in low compression regimes.

*Table 3.* Comparison with state-of-the-art pruning methods on ImageNet using ResNet50 and CIFAR-100 using ResNet56. For each level of FLOP reduction (%), we highlight the best results in bold. Δ Acc. on ImageNet considers Top1 accuracy.

| | Method | Δ Acc. | FLOPs (%) |
|---|---|---|---|
| | DECORE (Alwani et al., 2022) (CVPR, 2022) | (+) 0.16 | 13.45 |
| | SOSP (Nonnenmacher et al., 2022) (ICLR, 2022) | (+) 0.41 | 21.00 |
| | GKP-TMI (Zhong et al., 2022) (ICLR, 2022) | (-) 0.19 | 22.50 |
| | **CKA$_3$ (Ours)** | **(+) 1.11** | 22.64 |
| | SOSP (Nonnenmacher et al., 2022) (ICLR, 2022) | (+) 0.45 | 28.00 |
| | **CKA$_4$ (Ours)** | (+) 0.74 | **28.30** |
| | GKP-TMI (Zhong et al., 2022) (ICLR, 2022) | (-) 0.62 | 33.74 |
| ResNet50 | LPSR (Zhang & Liu, 2022) (SPL, 2022) | (-) 0.57 | 37.38 |
| on | **CKA$_6$ (Ours)** | (-) 0.18 | 39.62 |
| ImageNet | CLR-RNF (Lin et al., 2023) (TNNLS, 2023) | (-) 1.16 | 40.39 |
| | DECORE (Alwani et al., 2022) (CVPR, 2022) | (-) 1.57 | 42.30 |
| | HRank (Lin et al., 2020) (CVPR, 2020) | (-) 1.17 | 43.77 |
| | SOSP (Nonnenmacher et al., 2022) (ICLR, 2022) | (-) 0.94 | 45.00 |
| | WhiteBox (Zhang et al., 2023) (TNNLS, 2023) | **(-) 0.83** | 45.60 |
| | **CKA$_7$ (Ours)** | (-) 0.90 | 45.28 |
| | DECORE (Alwani et al., 2022) (CVPR, 2022) | **(-) 4.09** | 60.88 |
| | **CKA$_9$ (Ours) + $\ell_1$-norm** | (-) 5.15 | 62.00 |
| | DLRFC (He et al., 2022) (ECCV, 2022) | (+) 0.27 | 25.50 |
| | FRPP (Chen & Zhao, 2019) (TPAMI, 2019) | (-) 0.23 | 38.30 |
| | GCNP (Jiang et al., 2022) (IJCAI, 2022) | 0.00 | 48.77 |
| ResNet56 | EKG (Lee & Song, 2022) (ECCV, 2022) | (+) 0.31 | 50.00 |
| on | GCNP (Jiang et al., 2022) (IJCAI, 2022) | (-) 0.64 | 52.22 |
| CIFAR-100 | LPSR (Zhang & Liu, 2022) (SPL, 2022) | (-) 1.22 | 52.68 |
| | DAIS (Guan et al., 2023) (TNNLS, 2023) | **(+) 0.81** | 53.60 |
| | **CKA$_{17}$ (ours)** | (+) 0.71 | **63.79** |
| | **CKA$_{19}$ (ours)** | (-) 0.59 | 71.29 |
| | **CKA$_{20}$ (ours)** | (-) 1.96 | **75.05** |

As we mentioned before, our method is orthogonal to other pruning categories (i.e., the ones in Table 2); therefore, we can combine it with these techniques. Built upon previous ideas (Jordao et al., 2020), we take one of our pruned models and further prune it using the popular $\ell_1$-norm filter pruning. In this scenario, we achieve even better results, surpassing our best performance gains (using layer pruning only) in terms of FLOP reduction by 5.9 pp. Specifically, our method achieves a FLOP reduction above 80% while maintaining the accuracy drop below one pp. The single method paired with this level of reduction, DECORE (Alwani et al., 2022), exhibits an accuracy degradation of 2.41 pp compared to the original model.

We also evaluate our method on ImageNet and CIFAR-100 in Table 3. On these datasets, we observe a similar trend with the CIFAR-10 discussion when comparing our method against state-of-the-art pruning techniques. Particularly, on ImageNet, the layer-pruning approach by Zhang et al. (2022), LPSR, notably hurts the accuracy, whereas our method is capable of improving it while removing more FLOPs. It is important to mention that, due to technical details (see Appendix 6.1), layer-pruning methods are restricted to a set of candidate layers. In our experiments, we reach the limit of layer removal, reported in Table 3 (left). Therefore, we combine our method with the $\ell_1$-norm criterion to further prune our model in terms of filters, achieving a higher computational reduction of 62.00%. It is important

to note that more elaborate combinations could minimize the accuracy drop, but we leave this exploration for future work.

**Effectiveness in Shallow Architectures.** Among the keys to the success of pruning is the overparameterized regime of neural networks, particularly evident in deep models. Although our method is suitable in such cases, this experiment verifies its applicability to shallow models (i.e., ResNet32/44). Due to space constraints, we refer interested readers to Appendix 6.3 for detailed results.

On these architectures, our results align with previous experiments on deeper models (e.g., ResNet56/110), obtaining a satisfactory performance. Specifically, we remove 54.60% and 62.95% of FLOPs on ResNet32/44, respectively, without compromising accuracy. Beyond these levels of FLOP reduction, we observe a slight drop in accuracy, although it remains negligible (below 0.3 pp). Importantly, this behavior only occurs with deep networks when the FLOP reduction is above 70%. We also compare the performance of our layer-pruning method against state-of-the-art methods in these shallow architectures. Our method achieves competitive results, outperforming existing methods in most cases. We also evaluate our method on the MobileNetV2 and Transformer (see Appendices 6.3 and 6.4) architectures. On these architectures, we observe the same trend: our method removes notable FLOPs without hurting predictive ability.

Overall, the previous discussion confirms that our method is effective for both deep and shallow networks, as well as for other modern architectures.

**Robustness to Adversarial Samples.** Previous research have demonstrated the potential of pruning as a successful defense mechanism against adversarial attacks and out-of-distribution examples (Bair et al., 2024). In particular, evaluating pruned models in adversarial scenarios plays a critical role, as we need to guarantee the trustworthiness of these models before deploying them in real-world applications such as autonomous driving. To assess the adversarial robustness of the pruned models, we employ CIFAR-C (Hendrycks & Dietterich, 2019), ImageNet-C (Hendrycks & Dietterich, 2019) and CIFAR-10.2 (Lu et al., 2020).

On these datasets, our pruned models obtained superior robustness compared to the unpruned model. We notice a similar trend when evaluating the pruned models against the FGSM attack (see Figure 4 in supplementary material).

The previous discussion indicates that our method operated as a defense mechanism against adversarial attacks. Such a finding is unsurprising, as previous works have demonstrated the potential of pruning as a defense mechanism against adversarial attacks (Jordao et al., 2023). Conversely, Bair et al. (Bair et al., 2024) argued that pruning networks

hurt generalization on OOD. We observe that the pruned networks yielded by our method preserve OOD generalization. In this direction, Masarczyk et al. (2023) observed that some layers hinder OOD generalization. According to this experiment, our method mitigates this problem, even though it was not specifically designed for this purpose. Further analysis is required in this context, such as considering more attacks, but we leave it for future research.

**GreenAI and Transfer Learning.** The current consensus is that deeper models yield better predictive ability. A consequence of this paradigm is the computational overhead seen in modern architectures, contributing to an increase in energy demands, both in the training and deployment phases. According to previous works (Lacoste et al., 2019; Strubell et al., 2019; Faiz et al., 2024), these demands result in high carbon emissions ($CO_2$). Fortunately, the benefits in FLOP reduction and latency promoted by our method directly translate into a reduction of $CO_2$. For example, our best-pruned version of ResNet56 implies a reduction of $CO_2$ by $67.88\%$ during the fine-tuning. On ResNet110, our pruned model at the highest FLOP reduction regime leads to $80.85\%$ of $CO_2$ reduction. We can further evidence this practical reduction in transfer learning scenarios, where fine-tuning the models is necessary for downstream tasks. To do so, we employ the pruned versions of ResNet56 on CIFAR-100 and transfer their knowledge to CIFAR-10. Interestingly, we observe that our pruned model with the highest FLOP reduction achieved a $CO_2$ reduction of $68.23\%$ while maintaining accuracy within $1$ pp compared to the unpruned model. In addition, pruned models with lower FLOP reductions achieve better transfer learning results, corroborating the findings by Xu et al. (2023). Such behavior suggests a challenge for the current evaluation of pruning techniques: the quality of pruning should consider its performance in transfer learning tasks.

We believe the results above pose an important step towards Green AI. Particularly on the learning paradigm involving foundation models, as the success of this emerging field relies on transfer-learning (and self-supervised), hence, requiring fine-tuning (Bommasani et al., 2021; Evci et al., 2022; Amatriain et al., 2023).

## 5. Conclusions

Layer pruning emerges as an exciting compression and acceleration technique due to more pronounced benefits in FLOP reduction and latency speed-up than other forms of pruning. Despite achieving promising results, existing criteria for selecting layers fail to fully characterize the underlying properties of these structures. To mitigate this, we proposed a novel criterion for identifying unimportant layers. Our method leverages the Centered Kernel Alignment (CKA) similarity metric to select such layers from a set of candidates. Powered by CKA, we showed that similar representations between a dense (unpruned) network and its optimal pruning candidate indicate lower relative importance, thus capturing underlying properties exhibited by layers and preventing model collapse. Extensive experiments on standard benchmarks and architectures confirm the effectiveness of our method. Specifically, our method outperforms existing layer-pruning techniques in terms of both accuracy and FLOP reduction by a large margin. Compared to other state-of-the-art pruning methods, we obtain the best compromise between accuracy and FLOP reduction. Particularly, at high FLOP reduction levels all methods fail to preserve accuracy, whereas our method exhibits either an improvement or negligible drop. In addition, our pruned models exhibit robustness to adversarial samples and positive out-of-distribution generation. Finally, our work also poses an important step towards Green AI by reducing up to $80.85\%$ of carbon emissions required for training and fine-tuning modern architectures.

## Acknowledgments

The authors would like to thank grant #2023/11163-0, São Paulo Research Foundation (FAPESP), and grant #402734/2023-8, National Council for Scientific and Technological Development (CNPq). Anna H. Reali Costa would like to thank grant #312360/2023-1 CNPq. This study was financed in part by the Coordenação de Aperfeiçoamento de Pessoal de Nível Superior – Brasil (CAPES) – Finance Code 001.

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

## 6. Appendix

### 6.1. Technical Details Involving Layer Pruning

Let $\mathcal{N}$ be a network composed of $L$ layers. We can express the output of $\mathcal{N}$ as a set of $|L|$ transformations $f_i(.), i \in \{1, ..., |L|\}$. For the sake of simplicity, each $f_i$ consists of a series of convolution, batch normalization, and activation operations. In this definition, we obtain the network output ($y$) by forwarding the input data, denoted by $X$, through the sequential layers $f$, where the input of layer $i$ is the output of the previous layer $i - 1$; therefore, $y = f_{|L|}(...f_2(f_1(X)))$. This composes the idea behind plain networks (i.e., VGG and AlexNet).

In residual-like networks, we can express the output $y_i$ of layer $i$ in terms of the transformation $f_i$ and the output $y_{i-1}$ from the previous layer (see Figure 3 top). Formally, we write:

$$y_i = f_i(y_{i-1}) + y_{i-1}. \tag{2}$$

Equation 2 composes a residual module, where the rightmost part is named *identity-mapping shortcut* (or identity for short). From a theoretical perspective, pruning the $i$-th layer corresponds to letting $y_i = y_{i-1}$. From a technical perspective, pruning the $i$-th layer involves connecting the output of layer $i - 1$ to the input of layer $i + 1$. Due to the residual nature (skip connection), we could accomplish this by just zeroing out the weights of $f_i(y_{i-1})$; thus, eliminating its contribution in Equation 2. However, this process does not ensure practical speed-up without specialized hardware for sparse computing. Instead, after selecting a victim layer (let's say layer $i$), we obtain a pruned model according to the following process. First, we create a novel architecture without layer $i$, resulting in a model comprising only the surviving layers (the pruned model). Then, from the old architecture (unpruned model), we transfer the weights of the survival layers (we can apply this idea to a set of layers at once) to this novel architecture. Figure 3 (bottom) illustrates this process.

From the process above, we highlight that the layer-pruning process removes building blocks (i.e., a set of transformations, $f_i(.)$) a.k.a modules. Therefore, the pruning also eliminates the corresponding activations and normalization layers from the block. This is a common process in layer-pruning techniques (Zhang & Liu, 2022; Jordao et al., 2020; Zhou et al., 2022).

As a final note, the pruning process cannot remove all layers composing a model due to incompatible dimensions between the input and output tensor between stages (layers operating on representations in the same resolution). In particular, the pruning is unable to remove layers between stages. Thus, given an architecture of $k$ layers within a stage, the pruning can remove at most $k - 2$ layers within

this stage. Importantly, this is the reason why we are unable to remove all layers of a model. This analysis corresponds to ResNet architecture and can vary depending on the architecture design of the model.

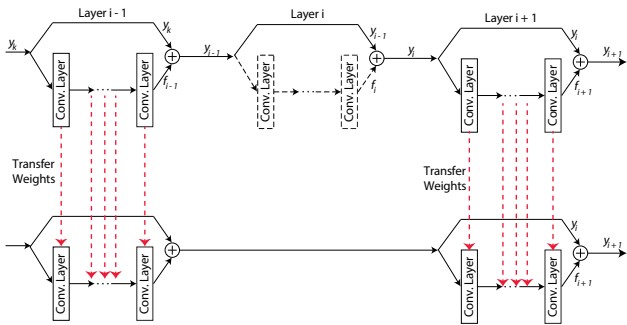

*Figure 3.* Architecture of a residual-like network. Top. The rationale behind this architecture is that the output of a layer takes into account the transformation performed by it ($f$) plus ($\oplus$) the input ($y$) it receives. Due to this essence, when we disable layer $i$ (its transformation – dashed lines), the output (representation) of layer $i - 1$ is propagated to layer $i + 1$, which means that the output $y_i$ belongs $y_{i-1}$. For the sake of simplicity, we omit the batch normalization and activation layers, which are also transferred in the process of layer removal. Bottom. Process to eliminate a layer from a technical perspective and, thus, obtain practical speed-up gains. After selecting the victim layer (i.e., Layer $i$), we create a novel architecture without it and, then, transfer the weights (red dashed arrows) of the corresponding survival layers.

## 6.2. Adversarial Attacks

Efforts toward a deeper understanding of the roles played by layers in network generalization and the effect of pruning in adversarial attack scenarios reinforce the idea that improving OOD and adversarial robustness while reducing computational demand is accomplishable for pruning methods (Masarczyk et al., 2023; Bair et al., 2024). Such compromises are crucial for deploying these models in safety-critical applications e.g., autonomous driving and robotics. In this experiment, we demonstrate that our pruned models improve OOD generalization and adversarial robustness. For this purpose, we compare the performance on different adversarial tasks of ResNet56 and its pruned models using our method. Figure 4 shows the results. According to this figure, our pruned models obtain positive robustness, highlighting their effectiveness even on severe compression rates. Specifically, on CIFAR-10.2 (Lu et al., 2020) and CIFAR-C (Hendrycks & Dietterich, 2019), we reduce more than 70% of computational cost while exhibiting improvements. Notably, only three pruned models obtained lower performance compared to the unpruned model, yet less than one pp. In the FGSM attack, regardless of the FLOP reduction levels, our pruned models dominate its unpruned version in

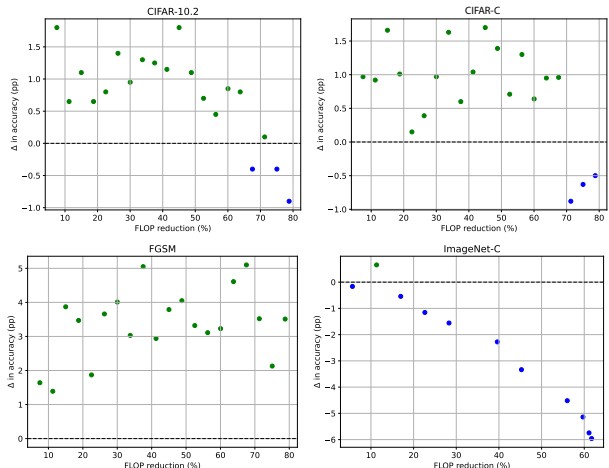

*Figure 4.* Results of pruned models for different adversarial attacks. Green and blue points correspond to an accuracy improvement and degradation, respectively. Dotted lines separate the plots into improvement and degradation groups. Top-Right: Results on out-of-distribution using CIFAR-10.2 (Lu et al., 2020). Top-Left: Results on adversarial robustness using CIFAR-C (Hendrycks & Dietterich, 2019). Bottom-Left: Results on FGSM adversarial attack. Bottom-Right: Results on ImageNet-C using pruned models from ResNet50

terms of adversarial robustness.

It is worth mentioning that during the pruning process, we avoid any defense mechanism. Therefore, the preceding discussion confirms that the benefits of our pruned models extend beyond computational gains.

## 6.3. Results on Shallow Architectures

In this experiment, we further explore the effectiveness of our method in shallow architectures: ResNet32 and ResNet44. We also consider the lightweight architecture MobileNetV2. Table 4 summarizes the results.

According to Table 4, our method removes up to 54.61% and 62.95% of FLOPs from residual architectures without compromising model accuracy. On higher FLOP reduction regimes, the performance drop is negligible (i.e., less than one pp.). This behavior is consistent with our analysis considering deeper models and supports the effectiveness of our method on shallow architectures.

Finally, our pruned models outperform state-of-the-art pruning methods; the only exception is the method by Nonnenmacher et al. (Nonnenmacher et al., 2022) on ResNet32. It turns out that our method reached the limit of layer removal (i.e., there are no more available layers to remove, see Section 6.1). Therefore, to achieve higher computational gains we should combine it with filter-pruning strategies. As a final note, the reason few methods appear in the table is that

*Table 4.* Comparison of state-of-the-art pruning methods on CIFAR-10 using ResNet32, ResNet44 and MobileNetV2. The symbols (+) and (-) denote increase and decrease in accuracy regarding the original (unpruned) network, respectively. For each level of FLOP reduction, we highlight the best results in bold.

| Method | | $\Delta$ Acc. | FLOPs (%) |
|---|---|---|---|
| ResNet32 | GKP-TMI (Zhong et al., 2022) (ICLR, 2022) | (+)0.22 | 43.10 |
| | SOKS (Liu et al., 2023) (TNNLS, 2023) | (-) 0.38 | 46.85 |
| | CKA (ours) | **(+) 0.68** | **47.78** |
| | DAIS (Guan et al., 2023) (TNNLS, 2023) | **(+) 0.57** | 53.90 |
| | SOKS (Liu et al., 2023) (TNNLS, 2023) | (-) 0.80 | 54.58 |
| | CKA (ours) | (+) 0.05 | **54.61** |
| | SOSP (Nonnenmacher et al., 2022) (ICLR, 2022) | (-) 0.24 | **67.36** |
| | CKA (ours) | **(-) 0.18** | 61.44 |
| ResNet44 | DCP-CAC (Chen et al., 2021) (TNNLS, 2022) | (-) 0.03 | 50.04 |
| | AGMC (Yu et al., 2021) (ICCV, 2021) | (-) 0.82 | 50.00 |
| | CKA (ours) | **(+) 0.47** | **53.27** |
| | CKA (ours) | (+) 0.22 | 62.95 |
| | CKA (ours) | (-) 0.29 | 72.64 |
| MobileNetV2 | CKA (ours) | (+) 0.17 | 26.60 |
| | CKA (ours) | (-) 0.37 | 31.28 |
| | CKA (ours) + $\ell_1$ | (-) 2.82 | 82.89 |
| | CKA (ours) + $\ell_1$ | (-) 3.44 | 85.06 |

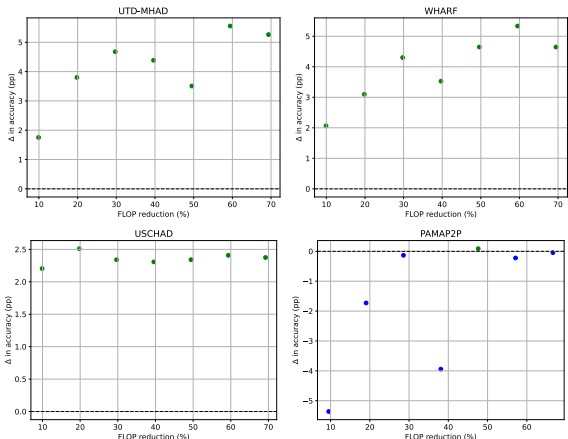

*Figure 5.* Performance of our layer-pruning method on Transformer architecture for human activity recognition based on wearable sensors (tabular data). Each point denotes a pruned model and the black-dashed line indicates the point where the drop in accuracy is zero; thus, points above this line (green) stand for pruned models with an improved accuracy compared to the original, unpruned, model.

there is a lack of pruning studies reporting results on these architectures (He & Xiao, 2023).

## 6.4. Results on the Transformer Architecture

Recent advancements in foundation models and general-purpose tasks often leverage Transformer architectures and their variants.

In the context of layer-pruning, Dong et al. (Dong et al., 2021) noted that Transformer-like architectures exhibit analogous behavior to ResNets – they experience either no degradation or only negligible drops in accuracy when removing certain layers. Such a claim provides a guarantee to perform layer-pruning on these architectures.

In this experiment, we evaluate the effectiveness of our layer-pruning method on the widely used Transformer architecture. Unfortunately, our limited computational budget prevents us from considering Visual Transformers, which typically require thousands of samples (e.g., JFT-300M) to achieve competitive results compared to convolutional networks. Thereby, we assess our layer-pruning technique in Transformers for human activity recognition based on wearable sensors, a popular application involving tabular data. Details about these datasets are available at this link: *https://doi.org/10.1016/j.neucom.2020.04.151*.

Our Transformer architecture (unpruned) comprises 10 layers, each with 128 heads and projection dimensions of 64. For each dataset, we train this Transformer architecture for 200 epochs and subsequently prune it similarly to the approach described in the main body of the paper. We emphasize that our objective here is not to advance the state-of-the-art; rather, we aim to demonstrate that the effectiveness

of our layer-pruning extends beyond ResNet architectures.

Figure 5 shows the results. In this figure, the black dashed line shows the point where the drop in accuracy is zero; hence, pruned models (green points) above this line exhibit an accuracy improvement. From Figure 5, we observe that most pruned models exhibited no accuracy drop, even on high compression regimes. Therefore, we conjecture that on these datasets the layer-pruning technique operated as a strong regularization mechanism. We observe that this regularization mechanism performs well as a function of the dataset size. For example, the datasets on the upper side of Figure 5 are at least 3 times smaller (in terms of training size) compared to the ones on the lower side. We intend to further explore this behavior in future research, particularly in low-data regimes, which is a well-known deficit in Transformer-like architectures. In summary, the results of this experiment confirm the effectiveness of our layer-pruning method on the Transformer architecture for tabular data.

