# OpenReview forum: "Effective Layer Pruning Through Similarity Metric Perspective"
_ICML.cc/2024/Workshop/WANT — WANT@ICML 2024 Poster_

### Official Review · Reviewer_sPhz · 2024-06-11

**Confidence:** 3

**Summary:**

This study investigates layer-pruning strategies and proposes an effective method, which estimates the importance of layers (building blocks) based on a Centered Kernel Alignment metric. The authors test the effectiveness of the method using a wide array of model architectures and benchmarks and find the proposed method is more robustness to aggressive compression rates, adversarial and OOD samples.

Pros:

- It demonstrates the effectiveness of the proposed methods across extensive experiments on standard architectures and benchmarks.
- Using a more practical metric, latency, to measure the model efficiency.

Cons:

- The authors only provide a brief description of the CKA criterion, the core of the proposed method, even without giving some intuitions why this similarity-based criterion can work.
- The proposed layer-pruning method seems not compatible with other filter-pruning methods, such as l1-norm, because it always leads to worse performance with l1-norm. Can you add test results of CKA + l1-norm for other levels of parameter reduction?

Comments:

- Can you show the evidence (e.g., ablation study or prior work) to support the motivation that layers contributing to similar representations are unimportant layers and can be removed?
- In Table 3, for ResNet50 on ImageNet, -0.83 (WhiteBox, 45.6% parameter reduction) should be in bold instead of CKA_7.

**Strengths:**

- extensive experiments on standard architectures and benchmarks.
- comprehensive evaluations, including robustness to OOD and adversarial attack

**Weaknesses:**

- lack direct evidence to the motivation
- lack novelty

---

### Official Review · Reviewer_VKxZ · 2024-06-11
**This paper revisits layer pruning granularity and proposes a new layer pruning algorithm using a novel criterion.**

**Confidence:** 4

**Summary:**

This paper revisits the layer pruning granularity and proposes a new layer pruning algorithm with the usage of a novel criterion called CKA. The author argues that this work is the first paper that brings CKA, most widely used for comparing network representations, into pruning literature. They show their proposed CKA method achieves better accuracy retention while reducing more FLOPs across different model-dataset pairs, and at the same time achieve better wall-clock speedup compared to methods in filter pruning granularity. They also show that their method can be used jointly with pruning methods in other granularities (CKA layer pruning + Lp-Norm filter pruning), and produce even better results..

Overall score: Borderline accept

**Strengths:**

Quality:
The storytelling and illustration of the algorithm is easy-to-follow.

Significance:
Revisit layer pruning granularity and bring it back to the pruning community with the discovery of its neglect potential.
First work considers CKA as a pruning criterion.

**Weaknesses:**

Creativity:
Revisiting layer pruning granularity back to recent literature can be interesting, but applying an existing criterion (even not in this field) to an existing pruning granularity is somewhat incremental.
The comprehensiveness of the experiment:
Although the reviewer admits that it is hard to make a hundred percent fair comparison between different pruning techniques in terms of the same:
1.Pretrain baseline model accuracy,
2.Training / fine-tuning pipeline setting,
3.Total cost / budget,
it is important to make everything (if not all) as transparent as possible.
The author does not report the pretrain model performance, training/fine-tuning budget (in terms of epoch), and the training/fine-tuning hyperparam, etc. While it is OK to compare the acc change between different methods, it is not fair if the pretrain model performance is somewhat weak or not comparable, or if the total budget is significantly different from other compared methods.
Lack of speed-test on pruning procedure

---

### Official Review · Reviewer_28X9 · 2024-06-12
**Using CKA as a similarity metric in iterative layer pruning leads to accurate and robust networks.**

**Confidence:** 4

**Summary:**

This paper explores the efficacy of using the Centered Kernel Alignment (CKA) metric (which closely borders standard correlation) as a means of measuring similarity between pruned and unpruned networks. In particular, the authors compute the CKA of the final hidden layer representation between a base and pruned network, $N$ and $N'$, where $N'$ consists of one less the layers in $N$.

In terms of the algorithm itself, the paper proposes a relatively simple strategy: 1) construct a set of candidate layers 2) iterate through the set and compare the representations with the base network 3) remove the layer corresponding to the lowest similarity metric ($1-CKA$) 4) repeat until the desired number of layers are removed. It is shown through ample experimental results that this method results in better performance across a range of reduction values, as well as generalizing to more robust datasets (eg. CIFAR-C).

**Strengths:**

- The paper is well-written, containing minimal errors and being considerably easy to follow.
- Tables and figures are clear and formatted well, making reading results easy.
- Despite its simplicity, the method is able to demonstrate strong results, outperforming many SOTA pruning techniques across a variety of metrics.
- The studies in adversarial robustness and latency in addition to standard accuracy and FLOP reduction make for a strong case for the paper's efficacy.

**Weaknesses:**

- Using CKA as a metric for pruning is not a novel idea, and has been explored in the past in similar algorithms [1]. This should, at the very least, be noted in the Related Works section.
- Figure 1. seems a little cherry-picked, as one could simply take the worst performing algorithm at every reduction level as a comparison point. For a stronger graph, I would suggest varying SOTA algorithms across reduction levels as the authors have done for their own method.



[1] Lachance, Alex, "Using Centered Kernel Alignment for Understanding and Pruning Neural Networks" (2022). Open Access Master's Theses. Paper 2283. https://digitalcommons.uri.edu/theses/2283

**Limitations:**

- There is a potentially significant resource use that is not noted in the paper. That is, generating every representation requires two passes through the entire training sample $X$, and every layer is pruned iteratively. In pruning literature, these traits are described as 'data-free' and 'iterative-pruning.' Some methods compared against have the advantage of being data-independent (that is, no passes through samples are required) and non-iterative (one-shot pruning). This marks a limitation of the proposed algorithm.

**Suggestions:**

Overall, this was a strong paper and I commend the authors for this. Some potential suggestions are:
- Add the paragraph from the appendix about candidate layers to make clear that not all layers can be removed.
- Adjust figure 1 to account for reduction %s across methods rather than singular points at a particular reduction level.

---

### Meta-Review · Area_Chair_MCpC · 2024-06-18

**Recommendation:** Accept (Poster)
**Confidence:** 4

**Metareview:**

This work proposes a novel saliency criterion for layer/depth pruning, CKA, and demonstrates its effectiveness on a range of standard networks and benchmarks. The reviewers note the following high-level strengths and weaknesses.

* (+) Reviewers commend the relative simplicity of the approach; despite the simplicity, the approach appears to work well on real-world networks and tasks.
* (+) One reviewer points out the additional ablations performed for evaluating the effect of the method on adversarial robustness and latency.
* (+) Clear writing and presentation.
* (-) Multiple reviewers point out the lack of novelty of CKA, which is well-known in the literature. A clear comparison to related work would be helpful.
* (-) Relatively weaker approach compared to single-shot (non-iterative) data-independent pruning approaches, which may be more attractive for larger networks such as LLMs.
* (-) No clear explanation of why CKA works better than existing saliency metrics.

Overall, I recommend acceptance (poster). Authors, please incorporate the suggested changes made by reviewers.

---

### Decision · Program_Chairs · 2024-06-18

**Decision:**

Accept (Poster)

**Comment:**

We thank the authors for their time and contribution to WANT and we are pleased to share that after the reviewing process the paper has been accepted. Congratulations! We encourage the authors to consider reviewers' feedback for the improvement of the camera-ready version. We hope to see you in person at the workshop and brainstorm on efficient training research together!